# From Discrete Complexes to Metal–Organic Layered Materials: Remarkable Hydrogen Bonding Frameworks

**DOI:** 10.3390/molecules25061353

**Published:** 2020-03-16

**Authors:** Carla Queirós, Ana M. G. Silva, Baltazar de Castro, Luís Cunha-Silva

**Affiliations:** LAQV/REQUIMTE & Department of Chemistry and Biochemistry, Faculty of Sciences, University of Porto, 4169-007 Porto, Portugal; cpaqueiros@gmail.com (C.Q.); ana.silva@fc.up.pt (A.M.G.S.); bcastro@fc.up.pt (B.d.C.)

**Keywords:** metal–organic framework, alkaline-earth metals, benzenedicarboxylate ligands, crystal structure, structural diversity, hydrogen bonding networks

## Abstract

A series of metal–organic coordination complexes based on alkaline-earth metal centers [Mg(II), Ca(II), and Ba(II)] and the ligand 5-aminoisophthalate (aip^2−^) revealed notable structural diversity, both in the materials’ dimensionality and in their hydrogen bonding networks: [Mg(H_2_O)_6_]∙[Mg_2_(Haip)(H_2_O)_10_]∙(Haip)∙3(aip)∙10(H_2_O) (**1**) and [Mg(aip)(phen)(H_2_O)_2_]∙(H_2_O) (**2**) were isolated as discrete complexes (0D); [Ca(aip)(H_2_O)_2_]∙(H_2_O) (**3**), [Ca(aip)(phen)(H_2_O)_2_]∙(phen)∙(H_2_O) (**4**), and [Ba_2_(aip)_2_(phen)_2_(H_2_O)_7_]∙2(phen)∙2(H_2_O) (**5**) revealed metal–organic chain (1D) structures, while the [Ba(aip)(H_2_O)] (**6**) showed a metal–organic layered (2D) arrangement. Furthermore, most of these metal–organic coordination materials revealed interesting thermal stability properties, being stable at temperatures up to 450 °C.

## 1. Introduction

Polyfunctional rigid aromatic carboxylate ligands are frequently used in the synthesis of new coordination polymers (CPs), frequently designated as metal–organic frameworks (MOFs) due to the interest in the production of new functional materials [1,2]. In the last years, an enormous number of works have been published using benzenecarboxylic acids [3,4], such as, 1,3,5-benzenetricarboxylic acid (H_3_BTC) [5,6], 1,3-benzenedicarboxylic acid (*m*-H_2_BDC) [7,8], 1,4-benzenedicarboxylic acid (*p*-H_2_BDC, terephthalic acid) [9,10], and their amino-substituted derivatives [11,12].

5-Aminoisophthalic acid (H_2_aip, Scheme 1) is becoming a ligand with increasing importance in materials science, since it can adopt several coordination modes, associated with the two carboxylate groups and the amino function [13]. Recently, several CPs based on H_2_aip derivatives have been prepared [14,15,16,17]; however, additional research of these materials is crucial, especially concerning post-functionalization procedures and the development of derivatives with different bases, with the main goal of enhancing their applicability in several technological areas, including gas adsorption and photoluminescence.

In the last decade, the use of alkaline-earths as metal centers in materials’ syntheses has become more common due to the development of novel synthetic methodologies, including mechanochemical and ultrasonical synthesis [18,19], that circumvent the limitations associated to their synthesis, like low solubility [20]. The use of anionic oxygen ligands is known to promote the syntheses of coordination polymers with different dimensionality. Therefore, several MOFs with alkaline-earth metals coordinated to carboxylate ligands have been reported [21,22,23]. Some of their properties include catalytic activity, luminescence, increased thermal stability, and biological potentialities. The most important disadvantages in the use of alkaline-earth metals are the unexpected geometries and coordination numbers of the final structures, as their bonding is not affected by ligand field stabilization [24,25] contrarily to transition or lanthanide metal ions. MOFs containing alkaline-earth metals can find application in numerous fields, such as superconduction, metallic conduction, ferroelectric, and catalysis and can present interesting architectures with appealing properties [1,20,26,27,28,29,30]. For the majority of the elements of group 2, some of those properties include aqueous solubility, non-toxicity, and lower cost, representing an important advantage relative to transition or lanthanide metal ions, especially for dyes/pigments industries [20].

Considering the lower cost associated to alkaline earths-based materials, several reports of their applicability in distinct technological areas [31,32,33,34,35], and our research interest in multifunctional MOF-based materials [17,36,37,38,39,40,41], a series of metal–organic coordination complexes based on Mg(II), Ca(II), and Ba(II) was prepared and structurally characterized. The materials revealed an interesting structural diversity including discrete complexes (0D), metal–organic chains (1D), and metal–organic layers (2D), as well as remarkable hydrogen bonding frameworks. The thermal stability was further investigated by thermogravimetric analysis (TGA).

## 2. Results and Discussion

### 2.1. Syntheses of the Complexes and Materials

The development and preparation of new metal–organic coordination materials based on alkaline earth (AE) metal ions and H_2_aip were performed using different synthetic protocols, including conventional heating and hydrothermal/solvothermal synthesis (the conditions used are summarized in Table 1). A series of AE-aip-based multidimensional metal–organic coordination complexes were isolated as crystalline materials suitable for single-crystal XRD analysis. Their solid-state structures were determined, confirming the formation of compounds with distinct dimensionalities: Discrete complexes (0D), [Mg(H_2_O)_6_]∙[Mg_2_(Haip)(H_2_O)_10_]∙(Haip)∙3(aip)∙10(H_2_O) (**1**) and [Mg(aip)(phen)(H_2_O)_2_]∙(H_2_O) (**2**), metal–organic chains (1D), [Ca(aip)(H_2_O)_2_]∙(H_2_O) (**3**), [Ca(aip)(phen)(H_2_O)_2_]∙(phen)∙(H_2_O) (**4**), and [Ba_2_(aip)_2_(phen)_2_(H_2_O)_7_]∙2(phen)∙2(H_2_O) (**5**), as well as a metal–organic layer (2D) [Ba(aip)(H_2_O)] (**6**), where *phen* is 1,10-phenanthroline. Additionally, all the materials were characterized by FT-IR spectroscopy and their thermal stability investigated by thermogravimetric analysis (TGA).

### 2.2. Discrete Complexes (0D)

The crystal structure of compounds **1** {[Mg(H_2_O)_6_]∙[Mg_2_(Haip)(H_2_O)_10_]∙3(Haip)∙(aip)·10(H_2_O)} and **2** {[Mg(aip)(phen)(H_2_O)_2_]_2_∙2(H_2_O)} revealed two unprecedented binuclear discrete (0D) complexes, with interesting structural features. Compound **1** was crystallized from a water solution in the monoclinic unit cell, with the structure solved and refined in space group *P*2_1_/c (see the experimental section). The asymmetric unit (*asu*) contains a mononuclear [Mg(H_2_O)_6_]^2+^ cation and a binuclear [Mg_2_(Haip)(H_2_O)_10_]^3+^ cationic complex, 1 partially deprotonated ligand (Haip^−^), and 3 fully deprotonated ligands (aip^2−^), as well as 10 crystallization water molecules (Appendix A, ESI). The three crystallographic-independent Mg^2+^ centers show distorted octahedral geometries being coordinated by six *O*-atoms with typical Mg−O distances and O−Mg−O angles (for details about these distances and angles, see Appendix A). Mg3 bonds to six water molecules, leading to the formation of the [Mg(H_2_O)_6_]^2+^ cation, while Mg1 and Mg2 coordinate to five water molecules and an *O*-atom of the carboxylate group of the same bridging Haip^−^ ligand originating the binuclear cationic complex [Mg_2_(Haip)(H_2_O)_10_]^3+^ (Figure 1a).

The packing arrangement of the [Mg(H_2_O)_6_]^2+^ cation and the binuclear cationic complex [Mg_2_(Haip)(H_2_O)_10_]^3+^ as well as the uncoordinated H_2_aip residues resembles a supramolecular layered-type structure viewed in the [1 1 0] direction of the unit cell (Figure 1b). In fact, there is an evident alternation between organic (Haip^−^ and aip^2−^ ligands) and inorganic (Mg^2+^ centers and water molecules) layers along the *b*-axis, resembling a hybrid layered material. As a consequence of the large number of crystallographic-independent coordinated (16) and uncoordinated (10) water molecules, as well as several Haip^−^ and aip^2−^ ligands, the extended crystal structure of 1 shows a remarkable network of non-covalent intermolecular interactions, namely π···π stacking, and weak (C‒H∙∙∙O) and strong (N−H∙∙∙O and O−H∙∙∙O) hydrogen bonds, ultimately leading to a 3D supramolecular network (Figure 1b; for details about the geometry of the strong hydrogen bonds, see Appendix A). Adjacent organic molecules (both Haip^−^ and aip^2−^ ligands) interact by π···π stacking between the phenyl rings (*C_g_*···*C_g_* distances ranging between 3.7941(2) and 3.9605(2) Å; where *C_g_* represents the gravity center of the ring), leading to the formation of supramolecular infinite chains in an anti-parallel arrangement (adjacent chains are rotated by ca. 180°; Figure 1c). However, besides the occurrence of some cooperative C−H∙∙∙O weak hydrogen bonds, the crystalline packing is mostly driven by an extending network of stronger N−H∙∙∙O and O−H∙∙∙O intermolecular interactions (light blue dashed lines in Figure 1; N−H∙∙∙O were not drawn). Remarkably, the cautious analysis of these interactions revealed around 80 distinct strong hydrogen bounds involving adjacent water molecules and/or organic ligands, which enforce the overall cohesion of the extended crystal structure (Appendix A). In particular, the large number of crystallographically independent water molecules are engaged in cooperative water-to-water hydrogen bonding interactions, leading to the formation of distinct water clusters, such as the (H_2_O)_22_ (depicted in Figure 1d). Although recent reports describing a variety of discrete water clusters (for example tetramers, pentamers, hexamers, decamers, or dodecamers) and some polymeric clusters (predominantly infinite chains) [42,43], the description of large water molecule aggregates are much more limited and, to the best of our knowledge, the (H_2_O)_22_ cluster found in this structure is unique.

Compound **2**, formulated as [Mg_2_(aip)_2_(phen)_2_(H_2_O)_4_]_2_·2(H_2_O), crystallized in a monoclinic unit cell and in the space group *P*2_1_/c. The *asu* contains four Mg^2+^ metal centers (Mg1, Mg2, Mg3, and Mg4), four deprotonated ligands (aip^2−^), four phen ligands, four lattice water molecules as well as two crystallization water molecules, forming two binuclear equivalent complexes (Figure 2a and Appendix A in ESI). The four crystallographic independent Mg^2+^ centers found in the *asu* show similar distorted octahedral geometries being coordinated at two *O*-atoms from two carboxylate groups and two water molecules (Mg−O distances ranging from 2.003 to 2.100 Å) and two *N*-atoms from a *phen* ligand (Mg−N distances between 2.193 to 2.230 Å). Concerning the internal angles involved in the Mg^2+^ coordination centers, the O−Mg−O angles are between 89.6 and 177.3°, N−Mg−O angles range from 84.39 to 97.50°, and the N−Mg−N angles values vary from 74.30° to 75.26° (for more details about these distances and angles, see Appendix A).

The packing arrangement of the binuclear complex **2** also reveals a layered supramolecular arrangement viewed in the [1 1 1] direction of the unit cell, creating an “upper” and “lower” layer of binuclear complex entities (Figure 2c). In comparison to 1, the distribution of the binuclear complexes seems more distorted. This distortion in the packing arrangement inhibits the π···π stacking since the phen molecules are displaced laterally to each other or actually in opposite directions. However, the crystal structure through N−H⋅⋅⋅O and O−H⋅⋅⋅O hydrogen bonds ultimately leads to a 3D supramolecular network (Figure 2d).

### 2.3. Metal–Organic Chains (1D)

The replacement of Mg^2+^ by Ca^2+^ and Ba^2+^ as metal centers (maintaining the organic ligand, H_2_aip) in the synthetic procedures originated new metal–organic chain (1D) materials, **3**, **4**, and **5**, using slightly different approaches. For instance, compound **3**, formulated as [Ca(aip)(H_2_O)_2_]∙H_2_O, was prepared by an adaptation of a published protocol [44], in which a temperature of 323 K was used. This value might be higher or lower than that used by the original authors, considering that the article only mentions that the solution was heated. The reaction time was increased in 16 h (64 h vs. 48 h) and the conjunction of these reactional conditions and probably the crystallization period (around two weeks at r.t.) originated a new 1D coordination polymer.

The material **3** was isolated from a water solution, crystallized in the triclinic system, and the crystal structure solved in the centrosymmetric space group *P*-1 (see the experimental section). The *asu* revealed two crystallographic-independent Ca^2+^ centers, three fully deprotonated ligands (aip^2−^), and four coordinated and two uncoordinated water molecules (Appendix A). The two Ca^2+^ centers (Ca1 and Ca2) are structurally identical, since both are six coordinated and show octahedral distorted coordination geometry (Figure 3a): Each one is coordinated by three carboxylate *O*-atoms of three crystallographic-equivalent aip^2−^ ligands, one *N*-atom from another organic ligand and two water molecules, {CaO_5_N}. The distorted octahedral geometry is clearly confirmed by the examination of the O−Ca−O and N−Ca−O internal angles: While the *cis* angles are found in the ranges 76.40(5)–103.72(4)° for Ca1 and 79.10(5)–101.43(5)° for Ca2, the *trans* angles range from 158.27(5) to 172.25(5)° and 160.92(5) to 175.79(5)° for Ca1 and Ca2, respectively (for details about these bond lengths and angles, see Appendix A in Appendix A).

Each crystallographic-independent aip^2−^ ligand coordinates to four equivalent Ca^2+^ centers via three carboxylate *O*-atoms and the amino *N*-atom (see Appendix A). The coordination behavior previously described for the metal centers and the organic ligands leads to the formation of an 1D metal–organic chain with the appearance of a ladder (Figure 3b), arranged along the *a*-axis of the unit cell. The smallest and largest Ca∙∙∙Ca distances imposed within the ladder are 4.8834(5) and 8.3175(9) Å, respectively. The adjacent ladders interact by strong N−H∙∙∙O hydrogen bonds originating 2D supramolecular structures (layers) extended in the a*b* plane of the unit cell (Figure 3c; for geometric details about the hydrogen bonds see Appendix A). These supramolecular layers further pack along the [0 0 1] direction of the unit cell. This packing arrangement is directed by an extensive N−H∙∙∙O and O−H∙∙∙O hydrogen bonding network, involving the amine and carboxylate groups of the aip^2−^ ligands, as well as the coordinated and uncoordinated water molecules, ultimately leading to a 3D supramolecular structure.

The utilization of co-ligands in the preparation of CPs or MOFs is a recognized strategy that is frequently applied to modify the material dimensionality or adjust its properties. Thus, the 1,10-phenatroline (*phen*) was used as co-ligand. A mixture of H_2_aip, Ca(OH)_2_, and *phen* was added to a 1:1 water/methanol mixture and warmed to 323 K for 3h. After a few weeks, brown crystals were obtained with a 37% yield. The structure determined by SCXRD revealed a 1D metal–organic chain (Figure 4), formulated as [Ca(aip)(phen)(H_2_O)_2_]∙(phen)∙(H_2_O) **(4)**, which was previously published in 2010 [45]. Compound **4** was prepared with some differences relative to the original protocol: (i) The initial dissolution period of the reagents was neglected, (ii) NH_3_ was not added, and (iii) the crystals were obtained in the same crystallization period without the need to use a water bath at 323 K. Therefore, this protocol is less energetically consuming and, in some degree, more environmentally friendly.

It was confirmed that the *asu* of **4** is composed by one Ca^2+^, one aip^2−^ ligand, one coordinated *phen*, two coordinated water molecules, one uncoordinated *phen*, and one crystallization water molecule (Appendix A in ESI). The Ca^2+^ center {CaO_4_N_2_} possesses a distorted octahedral geometry, with the Ca1−O distances ranging from 2.249 to 2.388 Å, and the Ca1−N distances from 2.501 to 2.534 Å (Figure 4a; detailed information concerning the bond lengths and angles of the Ca1 coordination center is summarized in Appendix A). The crystallographic-independent aip^2−^ ligand coordinates to two equivalent Ca^2+^ centers by one *O*-atom of each carboxylate group, establishing a 1D zigzag metal–organic chain through a bridging mode (Figure 4b). Both coordinated and uncoordinated *phen* molecules are positioned face-to-face, which allows weak π···π interactions (Figure 4c). Furthermore the occurrence of an extensive hydrogen bonding network (C−H⋅⋅⋅O, O−H⋅⋅⋅O, N−H⋅⋅⋅O and O−H⋅⋅⋅N) between the 1D coordination chains leads to the formation of 2D supramolecular layers (Figure 4d; geometric details concerning all the types of hydrogen bonds are listed in Appendix A), which are further extended into a 3D supramolecular arrangement through hydrogen bonds (Figure 4e).

The maintenance of the synthetic strategy used to prepare the previous metal–organic chain material, just replacing the Ca^2+^ metal center by Ba^2+^, was carried out, as an attempt to increase the dimensionality of the material. A mixture of H_2_aip, Ba(OH)_2_, and *phen* was added one-by-one to hot water and maintained for 68 h at 323 K (hydrothermal reaction), and dark pink crystals suitable for X-ray diffraction were obtained in a 15% yield (compound **5**).

The crystal structure of **5**, formulated as [Ba_2_(aip)_2_(phen)_2_(H_2_O)_7_]∙2(phen)∙2(H_2_O), was unveiled in the triclinic space group *P1*, and the *asu* revealed two crystallographic non-independent Ba^2+^ centers, two fully deprotonated ligands (aip^2−^), two coordinated *phen* molecules, seven coordinated water molecules, two solvation water molecules, and two uncoordinated *phen* molecules (Appendix A). The two Ba^2+^ centers (Ba1 and Ba2) are structurally distinct and share three bridging bonds by three *O*-atoms, imposing a distance from each other of 4.451 Å. Ba1 is octa-coordinated due to bonds with one carboxylate *O*-atom of a aip^2−^ ligand, two *N*-atoms from the co-ligand (*phen*), and four water molecules, {BaO_6_N_2_}, while the Ba2 is a hepta-coordinated center, as consequence of the bonds to one carboxylate *O*-atom of a aip^2−^ ligand, two *N*-atoms from the co-ligand (*phen*), and three water molecules, {BaO_5_N_2_} (Figure 5). The Ba1−O and Ba1−N distances range from 2.726(2) to 3.017(4) Å and from 2.915(5) to 2.937(5) Å, respectively, while the Ba2−O are between 2.823(2)–2.987(4) Å, and Ba2−N distances from 2.855(4) to 2.956(5) Å (the bond distances and angles involving the two Ba coordination centers are listed in Appendix A). The coordinated *phen* molecules in adjacent metal centers are disposed in opposite directions but parallel with respective neighboring uncoordinated *phen* molecules (Appendix A). This *phen*-*phen* arrangement leads to π···π stacking, which seems to be more effective for the *phen* molecules coordinated and closer to the Ba2 metal center since the two *phen* are equally positioned while those associated to Ba1 are slightly twisted from one another.

One of the crystallographic-independent aip^2−^ ligand coordinates to three Ba^2+^ centers (two chelating bidentate interaction and a simple bond to carboxylate *O*-atoms) while the other only coordinates to one Ba^2+^ via a carboxylate *O*-atoms (see Appendix A). This coordination mode between the Ba^2+^ metal centers and the organic ligands leads to the formation of a 1D metal–organic ladder chains (1D CP) running along the *a*-axis of the unit cell (Figure 5b). The supramolecular layers further pack along the [1 0 0] direction of the unit cell (Figure 5c), through an extensive hydrogen bonding network of type N−H∙∙∙O, O−H∙∙∙N, and O−H∙∙∙O, involving the amine and carboxylate groups of the aip^2−^ ligands, the *N*-atoms of the *phen* coordinated molecules as well as the coordinated and uncoordinated water and *phen* molecules, ultimately leading to a 3D supramolecular structure (detailed information about geometry of the hydrogen bonds can be found in Appendix A).

### 2.4. Metal–Organic Layers (2D)

The complex **6**, [Ba(aip)(H_2_O)], that revealed a metal–organic layered structure (2D CP), was prepared using a much more sustainable method than that reported by reported by Wu and co-workers [46]. Herein, the application of conventional heating (only at 50 °C) followed by self-assembly originated the desired materials, in contrast with the solvothermal process (at 150 °C) previously employed. The crystal structure of this compound was determined in the triclinic space group *P*-1 (see more details in the experimental section), with the respective *asu* composed of only one Ba^2+^ center, one deprotonated ligand (aip^2−^), and one coordinated water molecule (Appendix A). The Ba^2+^ center is coordinated by six carboxylate *O*-atoms belonging to four symmetry-related aip^2−^ anionic ligands, an *N*-atom from another anionic organic ligand, and one water molecule, {BaO_7_N} (Figure 6). The geometry of this eight-coordinated center resembles a dodecahedron, highly distorted, with Ba−O distances ranging from 2.6802 to 2.8815 Å, and the Ba−N distance slightly longer [2.9255 Å]. The evident distortion of the Ba coordination center is supported by the values of the O(N)−Ba−O(N) internal angles: The *cis* angles are found between 45.82 to 126.71° while the *trans* angles range from 143.23 to 167.88° (for details about these bond lengths and angles, see Appendix A).

The aip^2−^ ligands coordinate to five Ba^2+^ centers by the carboxylate *O*-atoms and the amino *N*-atom (see Appendix A), leading to the formation of 2D metal–organic layers extended in the [1 0 0] direction of the unit cell (Figure 6b). The adjacent layers close pack along the *a*-axis of the unit cell. This packing arrangement is reinforced by an extensive network of strong hydrogen bonds (N−H∙∙∙O and O−H∙∙∙O), involving the amine and carboxylate groups of the aip^2−^ ligands, as well as the coordinated water molecule, originating a 3D supramolecular structure (Figure 6c; see Appendix A about the hydrogen bonds’ geometry).

### 2.5. Vibrational Spectroscopy and Thermal Stability

The vibrational spectroscopy was studied using FTIR-ATR and the results obtained for the compounds 1–6 are shown in Figure 7 (selected region from 1800 to 400 cm^−1^). The spectra of all the compounds reveal characteristic bands in the region from 3600 to 3000 cm^−1^ related to the O−H and N−H stretching, with higher expression/intensity on the hydrated materials (not shown). Furthermore, the materials present a large number of bands in the 1700–1300 cm^−1^ region that can be associated to the stretching of C−H, C=C, C−O, and N−O and to the bending of N−H and C−O. On the other hand, the multiple bands presented in the 900–600 cm^−1^ region are related to C=C and C−H bending.

The thermal stability of the materials was studied using thermogravometric analysis TGA (Figure 8). From the TGA curves of discrete complexes **1** and **2** (black and grey line, respectively), it is possible to verify that the main difference is the higher decrease (slope) of the first mass loss for **1**, which is due to the large difference in water content (lattice and coordinated) in the Mg(II) complexes (26 vs. 6 water molecules in **1** and **2**, respectively). Another difference observed relates to the thermal stability of the complexes: **1** seems to be slightly more stable than **2** (stable up to ≈ 490 °C for **1** vs. ≈ 390 °C for **2**). This fact can be associated to the intermolecular interaction networks (*π*∙∙∙*π* stacking and H-bonds), which is more expressive in the extended structure of the complex **1**. For material **3**, two mass losses are observed, the first one (≈25%, calculated value of 20%) corresponding to the loss of the three water molecules (two coordinated and one uncoordinated) and the second corresponding to the material destruction at ≈ 570 °C. On the other hand, material **4** presents several mass losses; these losses can be attributed to the loss of uncoordinated and coordinated water (≈7%, calculated value of 8%) and *phen* (≈37%, calculated value of 30%, and 68%, calculated value of 63%, respectively) molecules up to the material destruction at ≈ 500 °C. For material **5**, the TGA curve profile presents a series of mass losses (Figure 8, green line). The first two mass losses correspond to the expulsion of the lattice and coordination water molecules (≈9%, calculated value of 8%) while the following correspond to the release of the *phen* organic ligands followed by the loss of the aip^2−^ ligands, leading to a steady destruction of the material around 480 °C. The thermal stability of 6 was also studied and revealed that the structure is stable up to 430 °C (Figure 8, wine line). The first mass loss of approximately 7% corresponds to the loss of the lattice water (calculated value of 5%) and the second one, ≈ 42% (expected value of 41%), corresponds to the loss of the ligand aip^2−^ and the corresponding destruction of the coordination polymer (CP). Most of the materials, with the exception of **4** and **5**, are stable up to at least 450 °C and their stability can be correlated to the corresponding structures and intermolecular interactions.

## 3. Experimental Section

### 3.1. Materials and Methods

Reagents and solvents were purchased as reagent-grade products and used without further purification unless otherwise stated. Microwave-assisted reactions were carried out in a CEM Discovery Labmate circular single-mode cavity instrument (300 W max magnetron power output) from CEM Corporation. Microanalyses were acquired by *Unidad De Análisis Elemental* of *Santiago de Compostela*. FT-IR spectra were obtained neat with a FT-IR Perkin Elmer Spectrum BX with an attenuated total reflectance (ATR) accessory.

### 3.2. Preparation of Materials

[Mg(H_2_O)_6_]∙[Mg_2_(Haip)(H_2_O)_10_]∙3(Haip)∙(aip)∙10(H_2_O) (**1**). H_2_aip (0.25 g, 1.38 mmol) and Mg(OH)_2_ (0.08 g, 1.37 mmol) were added to hot water (5 mL). The mixture was maintained for 65 h at 323 K, and after cooling was filtered and placed on a 30-mL glass vial. Red/brown crystals were obtained after a few weeks in 12% yield (243 mg). Elemental analysis (%) calcd. for C_40_H_81_Mg_3_N_5_O_46_ (1441.03 g/mol): C 33.34, H 5.67, N 4.86; found: C 33.33, H 5.92, N 4.43; Selected FT-IR (solid phase, *ν*/cm^−1^): 3200 m, 2944 w, 2842 w, 2362 m, 2342 m, 1556 s, 1466 w, 1412 m, 1354 s, 916 w, 890 w, 700 w.

[Mg(aip)(phen)(H_2_O)_2_]∙(H_2_O) (**2**). H2aip (1.0 equiv., 1.00 mmol, 182.1 mg), Mg(OH)2 (1.34 equiv., 1.34 mmol, 78.3 mg) and 1,10-phen (0.62 equiv., 0.62 mmol, 122.0 mg) were added to a 1:1 mixture of H_2_O/acetone (10 mL). The mixture was transferred to a 23-mL Teflon-lined reactor. The reactor was sealed and heated to 403 K for 72 h. After cooling, pink/brown crystals were collected by filtration and dried at 323 K for 5 h and a 45% yield was obtained (385.3 mg). Selected FT-IR (solid phase, *ν*/cm^−1^): 3242 m, 2355 m, 1668 w, 1610 m, 1540 s, 1444 m, 1472 m, 1375 s, 1006 w, 980 w, 922 m, 770 m, 709 m.

[Ca(aip)(H_2_O)_2_]∙(H_2_O) (**3**). A mixture of H_2_aip (0.50 g, 2.76 mmol) and Ca(OH)_2_ (0.21 g, 2.83 mmol) was prepared in water (10 mL) and heated at 323 K during 64 h. The mixture was filtered, and the filtrate was transferred to a 20-mL glass vial and kept at room temperature for several days, after which a clear pink crystalline solid began to form on the vial walls, in 28% yield (208 mg). Elemental analysis (%) calcd. for C_8_H_11_CaNO_7_ (273.26 g/mol): C 35.16, H 4.06, N 5.13; found: C 35.18, H 4.05, N 5.53. Selected FT-IR (solid phase, *ν*/cm^−1^): 3385 s, 3313 w, 3203 w, 1556 m, 1541 m, 1516 s, 1471 s, 1425 m, 1375 m, 1139 s, 998 s, 845 m, 780 m, 724 m, 636 s.

[Ca(aip)(phen)(H_2_O)_2_]∙(phen)∙(H_2_O) (**4**). H2aip (1.0 equiv., 1.00 mmol, 97.8 mg), Ca(OH)_2_ (1.0 equiv., 0.57 mmol, 42.5 mg) and water (5 mL) were placed in a 10-mL reaction vessel, which was then closed and placed in the cavity of a CEM microwave reactor. The reaction vessel was irradiated at 383 K for 4 h. After cooling, the mixture was filtered and transferred to a glass vessel and was allowed to stand for 24 h after which a pale pink crystalline solid began to deposit on the vessel bottom. Single crystals suitable for X-ray diffraction were collected with 83% yield (138.2 mg). Selected FT-IR (solid phase, *ν*/cm^−1^): 3250 w, 1644 m, 1610 s, 1538 m, 1470 s, 1444 s, 1390 m, 1324 s, 1276 s, 892 s, 780 s, 734 s.

[Ba_2_(aip)_2_(phen)_2_(H_2_O)_7_]∙2(phen)∙2(H_2_O) (**5**). A conventional heating procedure was utilized in the preparation of the material: H_2_aip (1.0 equiv., 2.76 mmol, 500.2 mg), Ba(OH)_2_ (0.90 equiv., 2.48 mmol, 425.0 mg) and 1,10-phen (0.60 equiv., 1.66 mmol, 338.2 mg) were added one-by-one to hot water (10 mL). The mixture was maintained at 323 K for 68 h. After cooling, the mixture was filtered and the solution concentrated in a rotavapor until some precipitation occurred. The mixture was then transferred to a 30-mL glass vessel, and chloroform and methanol were added. Dark pink crystals suitable for X-ray diffraction were obtained after a month with 15% yield (371.3 mg). Selected FT-IR (solid phase, *ν*/cm^−1^): 3376 m, 3314 m, 2941 w, 2358 w, 2338 w, 1542 s, 1469 m, 1424 m, 1374 s, 1320 m, 894w, 826 w, 785 m, 774 m, 719 m

[Ba(aip)(H_2_O)] (**6**). H_2_aip (0.25 g, 1.38 mmol), Ba(OH)_2_ (0.21 g, 1.22 mmol) were added to hot water (5 mL). The mixture was maintained for 65 h at 323 K. After cooling, the mixture was filtered and the solution was kept at room temperature for a month. Dark pink crystals were collected and dried in air with 59% yield (242 mg). Elemental analysis (%) calcd. for C_8_H_7_BaNO_5_ (334.47 g/mol): C 28.73, H 2.11, N 4.19; found: C 28.70, H 2.14, N 4.09. Selected FT-IR (solid phase, *ν*/cm^−1^): 2944 m, 2367 m, 2346 w, 1554 m, 1436 w, 1353 m, 1320 m, 772 w, 718 w.

### 3.3. X-ray Diffraction

Crystalline material of complexes **1**–**6** were manually harvested from the respective crystallization vial, immersed in highly viscous inert oil, and a suitable single-crystal of each complex was mounted on a cryoloop or thin glass fiber [47]. Data were collected at 150 or 180 K on a Bruker X8 APEX II charge-coupled device (CCD) area-detector diffractometer (Mo K*_α_* graphite-monochromated radiation, *λ* = 0.71073 Å), controlled by the APEX2 software package [48]. Images were processed using the software package SAINT+ [49], and data were corrected for absorption by the multi-scan method implemented in SADABS [50]. The structures were solved by direct methods SHELXS-97 [51,52], permitting the direct location of most of the heaviest atoms, and the remaining non-hydrogen atoms were located from difference Fourier maps calculated from successive full-matrix least squares refinement cycles on *F*^2^ using SHELXL-97 [51,53].

All non-hydrogen atoms were successfully refined using anisotropic displacement parameters, and hydrogen atoms attached to carbon were set at idealized positions, and included in subsequent refinement cycles in riding-motion approximation with isotropic thermal displacements parameters (*U_iso_*) fixed at 1.2 × *U_eq_* of the carbon atom to which they are connected. In general, hydrogen atoms associated with the water molecules and nitrogen atoms were located in difference Fourier maps, the corresponding N–H, O–H, and H∙∙∙H distances were fixed at appropriated values, and included in subsequent refinement cycles in riding-motion approximation, with isotropic thermal displacements parameters (*U*_iso_) fixed at 1.5 × *U_eq_* of the respective parent atom. Table 2 summarizes the acquisitions data and structure refinements details for complexes **1**–**6**.

## 4. Concluding Remarks

The application of distinct synthetic protocols, namely conventional heating and hydrothermal/solvothermal synthesis and selected experiments, allowed the preparation of six crystalline metal–organic materials based on alkaline earth (AE) metal ions and H_2_aip. Their solid-state structures were unequivocally determined by single-crystal XRD, confirming the formation of coordination materials with distinct topologies and dimensionalities: Discrete complexes (0D), metal–organic chains (1D), and metal–organic layers (2D). The use of Mg^2+^ as the metal center originated the discrete complexes **1** and **2**, formulated as [Mg(H_2_O)_6_]∙[Mg_2_(Haip)(H_2_O)_10_]∙(Haip)∙3(aip)∙10(H_2_O) and [Mg(aip)(phen)(H_2_O)_2_]∙(H_2_O), respectively. The two materials prepared with Ca^2+^, [Ca(aip)(H_2_O)_2_]∙(H_2_O) (**3**), and [Ca(aip)(phen)(H_2_O)_2_]∙(phen)∙(H_2_O) (**4**) revealed 1D coordination polymeric structures. Interestingly, the Ba^2+^-based materials showed distinct coordination polymer dimensionalities: While **5**, [Ba_2_(aip)_2_(phen)_2_(H_2_O)_7_]∙2(phen)∙2(H_2_O), is a 1D coordination chain, **6**, [Ba(aip)(H_2_O)], is a 2D coordination layer. All six materials comprise coordinated and/or uncoordinated water molecules, which are engaged in extensive networks of hydrogen bonds. The “hydration level” of the materials seems to influence their thermal stability, however, some of these materials were found to be stable up to 450 °C. Furthermore, the structures reported reveal the importance of the coordination behaviors of 5-aminosiophthate, which is useful for developing functional coordination network materials that take advantage of the amino functionality by post-synthetic modifications. Actually, this research work has been extended to new generations of functional MOF materials applied as photoluminescent sensors and oxidative catalysts.

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
