# Peer review of "From Discrete Complexes to Metal–Organic Layered Materials: Remarkable Hydrogen Bonding Frameworks"

_molecules, 2020, doi:10.3390/molecules25061353_

Round 1

Reviewer 1 Report

After reading the assigned manuscript by Queirós et al., I consulted the Aims and Scope of the journal. The opening sentence of the Aims of the journal states "Our aim is to provide rigorous peer review and enable rapid publication of cutting-edge research to educate and inspire the scientific community worldwide."  My assessment of this manuscript is that it is a series of structure reports that are linked by a common ligand and group of elements (the alkali earths).  Beyond reporting these structures the manuscript fails in the ideal of educating and inspiring the community of chemists working on coordination polymers or MOFs. 

The introduction is mostly well written and identifies areas lacking in the literature concerning materials of the type reported in the paper: "additional research ..... concerning post-functionalization procedures and the development of derivatives with different bases, with the main goal of enhancing their applicability in several technological areas, including gas adsorption and photoluminescence.  Unfortunately the manuscript does not tackle any of these areas.  The introduction appears however to cover off the main areas related to the field and is a readable and technically correct presentation of the literature.  The terminology used to describe these discrete complexes and coordination polymers is a little confusing - they are referred to as Metal-Organic Layered Materials, coordination polymers, hydrogen bonded frameworks, and metal-organic coordination complexes (which is a tautology).

The experimental is reasonably clear in so far as these types of structure reports are concerned. A procedure is noted for all materials prepared but given the vast scope for materials made from the title ligand, 1,10-phenanthroline and alkali earth metals if would be unlikely if this was the a full account of the reaction space. Some of the materials lack combustion analysis data for composition and I did not note PXRD data to attest to phase purity of the samples.

The rationalization of the experimental conditions (reported in the results and discussion) is weak, and does not provide any useful insights. For example, there is a previous report (cited in the manuscript - Inorg Chem 2007, 46, 6828–6830) which describes a helical Ca structure from a similar combination of reagents.  Aside from a brief acknowledgement, no studies were conducted to understand how the synthesis of these two materials can be controlled nor the relationship between these materials - these types of insights are useful to the community struggling under the weight of a large number of materials/structures.

The remainder of the paper is a series of structure descriptions that are clear and serve the purpose of describing the structures; however, they provide very limited connection to the broader literature.  It would help if the topology of the structures were included.

In summary, I do not recommend the manuscript for publication in its current form. The major modifications I recommend ahead of a resubmission are a more comprehensive study of the synthesis of these materials, particularly to understand the different phases available from similar combinations of the materials (e.g. for the Ca structure), and ideally some investigations that fill the gaps in the literature noted in their introduction (e.g. a property utilising the amine in the linker).  I would suggest the remainder of the ad-hoc collection of structures be deposited in the CSD.  

Author Response

 Reviewer 1

After reading the assigned manuscript by Queirós et al., I consulted the Aims and Scope of the journal. The opening sentence of the Aims of the journal states "Our aim is to provide rigorous peer review and enable rapid publication of cutting-edge research to educate and inspire the scientific community worldwide." My assessment of this manuscript is that it is a series of structure reports that are linked by a common ligand and group of elements (the alkali earths). Beyond reporting these structures the manuscript fails in the ideal of educating and inspiring the community of chemists working on coordination polymers or MOFs.

The introduction is mostly well written and identifies areas lacking in the literature concerning materials of the type reported in the paper: "additional research concerning post-functionalization procedures and the development of derivatives with different bases, with the main goal of enhancing their applicability in several technological areas, including gas adsorption and photoluminescence.  Unfortunately the manuscript does not tackle any of these areas. The introduction appears however to cover off the main areas related to the field and is a readable and technically correct presentation of the literature.

Response:

The authors acknowledge the reviewer for this relevant comment however we would like to emphasise that this research work is only our first study concerning the coordination behaviour of this ligand (5-aminoisophtalate, aip) and their potentiality in the preparation of coordination polymers and/or metal-organic frameworks. In fact we consider that this manuscript represent a “kick-off” for the investigation, development and optimization of the preparation of a new generation of functional metal-organic framework materials based in this peculiar ligand. The reported and discussed results support the feasibility to obtain coordination polymers with the aip ligand, leaving the amino amino functionality for post-synthetic modifications. A sentence related with this relevant feature of the work was incorporated in the “concluding remarks” section.

Actually, this research work was extended to two new generations of functional metal-organic framework materials based in the common aip ligand and lanthanides (Eu3+, Tb3+, and others) or Cobalt (Co2+) as metal centres. While the first series of compounds was applied as photoluminescent materials, the second one was used in oxidative catalysis; these results will be reported in two future papers.

We believe this manuscript is effectively adequate and deserve the publication in these special issue dedicated to "Functional Metal-Organic Framework Based Materials".

The terminology used to describe these discrete complexes and coordination polymers is a little confusing - they are referred to as Metal-Organic Layered Materials, coordination polymers, hydrogen bonded frameworks, and metal-organic coordination complexes (which is a tautology).

Response: In fact, despite the term hydrogen bonded frameworks refer to a distinct type of structural arrangement we agree that the remaining terminology (namely metal-organic layered materials, coordination polymers and metal-organic coordination complexes) can be a bit confused. In order to mitigate this redundancy of terms (tautology) the text of the manuscript was revised accordingly.

The experimental is reasonably clear in so far as these types of structure reports are concerned. A procedure is noted for all materials prepared but given the vast scope for materials made from the title ligand, 1,10-phenanthroline and alkali earth metals if would be unlikely if this was the a full account of the reaction space. Some of the materials lack combustion analysis data for composition and I did not note PXRD data to attest to phase purity of the samples.

Response: we recognize that this comment of the reviewer is pertinent in the context of preparation and development of metal-organic framework materials to be directly applied in gas adsorption, catalysis, sensing and others. However, as mentioned previously this preliminary work was essentially focused in a structural study of the coordination behaviours of the ligand 5-aminosiophthate for developing coordination network materials, in particular the availability of the amino functionality for post-synthetic modifications. Consequently, we believe that the determination of solid-sate structures by single-crystal X-ray diffraction (complemented with vibrational spectroscopy and thermogravimetric analysis) is much more relevant than other characterization techniques, which are not indispensable for the objectives of this manuscript.

The rationalization of the experimental conditions (reported in the results and discussion) is weak, and does not provide any useful insights. For example, there is a previous report (cited in the manuscript - Inorg Chem 2007, 46, 6828–6830) which describes a helical Ca structure from a similar combination of reagents.  Aside from a brief acknowledgement, no studies were conducted to understand how the synthesis of these two materials can be controlled nor the relationship between these materials - these types of insights are useful to the community struggling under the weight of a large number of materials/structures.

Response: the authors recognize that the rationalization of the experimental condition, as well as the possibility to clear up the connection synthesis – structure – properties is always desirable in investigaions related with metal-organic framework compounds. However, we consider this is not the main objective of this preliminary manuscript, and requires another type of strategy (for example the use of only one type of metal, the ligand 5-aminosiophthate, and a systematic changes of the remaining parameters) and considerable amount of work, which deserves an investigation additional to be reported in other(s) publication(s).

The remainder of the paper is a series of structure descriptions that are clear and serve the purpose of describing the structures; however, they provide very limited connection to the broader literature.  It would help if the topology of the structures were included.

Response: following this suggestion, the topology of the structure of the metal-organic layered material (comp 6) was included in the manuscript, particularly an image of the topology in Figure 6. Naturally, the legend of this figure was modified accordingly.

In summary, I do not recommend the manuscript for publication in its current form. The major modifications I recommend ahead of a resubmission are a more comprehensive study of the synthesis of these materials, particularly to understand the different phases available from similar combinations of the materials (e.g. for the Ca structure), and ideally some investigations that fill the gaps in the literature noted in their introduction (e.g. a property utilising the amine in the linker). I would suggest the remainder of the ad-hoc collection of structures be deposited in the CSD.

Response: in fact, the final CIF files have been already deposited in the Cambridge Crystalographic Data Centre (CCDC), preceding the submission of this manuscript, the CCDC deposition numbers were now included in the Table 1 of the revised manuscript.

Reviewer 2 Report

This manuscript describe the synthesis and structural diversity of a series of metal-organic coordination complexes or coordination polymers (CPs) based on alkaline-earth metal centers [Mg(II), Ca(II) and Ba(II)] and the ligand 5-aminoisophthalate (aip2-). The synthesis of this work is quite routines and does not present any exciting features or interesting properties. There are some comments.

In this study, structural characterization of these compounds is the main work. However, the checkcif files for the six compounds are missing, especially for compound 5 with P 1 crystal system. It must be checked carefully. In compound 1, the figure caption of Fig. 1(d) is wrong, no N−H∙∙∙O hydrogen bond). Page 3, line 105-107, “the crystalline packing is mostly driven by an extending network of stronger N−H∙∙∙O and O−H∙∙∙O intermolecular interactions (light blue dashed lines in Figure 2).” should be checked and corrected. Figure 2 is for compound 2. For compound 3, Why the extended layered supramolecular arrangement and 3D supramolecular network are not shown in Fig. 2? About the thermal stability by TGA, the structural variation is important after the losses of guest molecules. In-situ powder x-ray diffraction measurement may be suitable on the explanation of the structural stability.

For these reasons, I didn’t see the importance of this work that qualified for publication on Molecules, which is one of international journals with high reputation.

Author Response

  Reviewer 2

This manuscript describe the synthesis and structural diversity of a series of metal-organic coordination complexes or coordination polymers (CPs) based on alkaline-earth metal centers [Mg(II), Ca(II) and Ba(II)] and the ligand 5-aminoisophthalate (aip2-). The synthesis of this work is quite routines and does not present any exciting features or interesting properties. There are some comments. In this study, structural characterization of these compounds is the main work. However, the checkcif files for the six compounds are missing, especially for compound 5 with P 1 crystal system. It must be checked carefully.

Response: the six structures have been checked carefully, the checkCIF files generated for all of them, and the final CIF files deposited in the Cambridge Crystalographic Data Centre (CCDC), preceding the submission of this manuscript. Following this suggestion the checkCIF files were now included in the SI and the CCDC deposition numbers incorporated in the Table 1 of the revised manuscript.

In compound 1, the figure caption of Fig. 1(d) is wrong, no N−H∙∙∙O hydrogen bond).

Response: the caption of the Figure 1, namely 1(d), was corrected.

Page 3, line 105-107, “the crystalline packing is mostly driven by an extending network of stronger N−H∙∙∙O and O−H∙∙∙O intermolecular interactions (light blue dashed lines in Figure 2).” should be checked and corrected. Figure 2 is for compound 2.

Response: The sentence was corrected.

For compound 3, Why the extended layered supramolecular arrangement and 3D supramolecular network are not shown in Fig. 2?

Response: following this pertinent question, the representations of the extended supramolecular arrangement and the 3D network were included in the manuscript, as part of the Figure 2, namely Figure 2c and 2d respectively. Furthermore, the legend of this figure and the text were adjusted.

About the thermal stability by TGA, the structural variation is important after the losses of guest molecules. In-situ powder x-ray diffraction measurement may be suitable on the explanation of the structural stability.

Response: as referred in a previous answer to a comment of reviewer 1, this preliminary work was essentially focused in a structural study of the coordination behaviours of the ligand 5-aminosiophthate in the preparation and development of coordination network materials, in particular the availability of the amino functionality for post-synthetic modifications. Consequently, we believe that the determination of solid-sate structures by single-crystal X-ray diffraction is much more relevant than other characterization techniques, which are not indispensable for the objectives of this manuscript. The TGA analysis reported for the six reported compounds are appropriated to verify that coordination compounds based in this ligand reveal enough thermal stability to be post-synthetic modified towards the preparation of functional materials. We recognize that in-situ powder X-ray diffraction measurements could be important for the characterization of functional materials, but considerably out of the scope of this manuscript.

Reviewer 3 Report

In this work, six coordination compounds with various dimensionality are prepared, and their crystal structures are determined. The used ligand is 5-aminoisophtalic acid and the metal ions are Mg(II), Ca(II), and Ba(II). All the crystals contain lattice water or coordinated water molecules that contribute to the formation of complicated hydrogen bonding. In addition, the amine functional group of the organic ligand also form N-H…O hydrogen bonding in all cases. Besides the hydrogen bond networks, the authors also found the pi-pi interactions among the 5-aminoisophthalate ligands, leading to characteristic supramolecular architectures. Importantly, this work reveals the unique coordination behaviors of 5-aminosiophthate, which is useful for developing functional coordination network materials because the amino functionality is very useful for post-synthetic modifications. In this regard, I would like to support the publication of this work.

Author Response

  Reviewer 3

In this work, six coordination compounds with various dimensionality are prepared, and their crystal structures are determined. The used ligand is 5-aminoisophtalic acid and the metal ions are Mg(II), Ca(II), and Ba(II). All the crystals contain lattice water or coordinated water molecules that contribute to the formation of complicated hydrogen bonding. In addition, the amine functional group of the organic ligand also form N-H…O hydrogen bonding in all cases. Besides the hydrogen bond networks, the authors also found the pi-pi interactions among the 5-aminoisophthalate ligands, leading to characteristic supramolecular architectures. Importantly, this work reveals the unique coordination behaviors of 5-aminosiophthate, which is useful for developing functional coordination network materials because the amino functionality is very useful for post-synthetic modifications. In this regard, I would like to support the publication of this work

Response: the authors are grateful by the positive comments and support of the Reviewer 3 concerning this manuscript.

Round 2

Reviewer 1 Report

The authors have made a few minor modifications to the manuscript but do not address the majority of the significant concerns raised in the first round of review. My major concern on the scientific importance of this contribution has not been assuaged and hence I am not changing my decision on the manuscript overall. 

Regardless of the editorial decision on the above point of difference, the authors do need to provide some evidence of compound purity for [Mg(aip)(phen)(H2O)2]∙(H2O), [Ca(aip)(phen)(H2O)2]∙(phen)∙(H2O) and [Ba2(aip)2(phen)2(H2O)7]∙2(phen)∙2(H2O).  According to their experimental details there is no shortage of material.  PXRD would provide the most robust determination that there is a single crystalline phase here.  I agree that the TGA data provides useful insights but PXRD would provide more compelling evidence.

Furthermore, at least some consideration of the reaction space that links the Ca structures reported herein to the previous structure in Inorg Chem 2007, 46, 6828–6830 would provide some context and purpose.

Reviewer 2 Report

The revised manuscript is corrected mostly based on reviewers' comments. I am wondered that the space group of compound 5 is P 1 or P -1. It should be carefully checked.